# Level and determinants of pentavalent vaccine dropout during infancy: A hierarchical analysis of community-level longitudinal study

Tariku Tesfaye Bekuma[ID][1,2]*, Assefa Seme[3], Saifuddin Ahmed[4], Muluemebet Abera[2]

1 Department of Population and Family Health, Institute of Public Health, Jimma University, Ethiopia, 2 School of Public Health, Institute of Health Sciences, Wallaga University, Ethiopia, 3 School of Public Health, College of Health Sciences, Addis Ababa University, Ethiopia, 4 Department of Population, Family and Reproductive Health, Johns Hopkins Bloomberg School of Public Health, Baltimore, Maryland, United States of America

* tarii2007@gmail.com

## Abstract

### Background

An estimated 23 million children under the age of one do not get the recommended basic vaccinations globally, and Ethiopia is one of the 10 countries where almost 60% of these children live. The alarming decline in childhood vaccination is leaving millions of children at risk of devastating but preventable diseases. Vaccination dropout poses a significant challenge in ensuring that children receive the full protection provided by vaccines. In Ethiopia, individual vaccine coverage has improved, but the proportion of fully vaccinated children with all recommended vaccines remains low. The study aimed to examine the level and determinants of pentavalent vaccine dropouts during infancy.

### Methods

The study utilized data from the PMA Ethiopia panel survey, which was conducted in three regions: Amhara, Oromia, and the former SNNP. A multi-stage cluster design with urban-rural stratification was used to draw a probability sample of households and women of reproductive age. We employed a multilevel logistic regression analysis in light of the hierarchically clustered nature of PMA data and the results are reported in odds ratio. The measures of variation were explained by intra-cluster correlation, median odds ratio, and proportional change in variance.

### Results

A total of 1295 infants were included in the final analysis. The result shows that a dropout from the first to the third dose of pentavalent was 37.48%. Pentavalent drop-out rates were about two times (AOR = 1.84, 95% CI: 1.24–2.72) higher for infants born at home than for babies born at a health facility, and two times (AOR = 1.77,

**Data availability statement:** The data underlying the results presented in the study are available on PMA Datalab at http://www.pmadata.org.

**Funding:** The author(s) received no specific funding for this work.

**Competing interests:** The authors have declared that no competing interests exist.

**Abbreviation: BMGF**, Bill and Melinda Gates Foundation; **DHS**, Demographic and Health Survey; **JHBSPH**, Johns Hopkins Bloomberg School of Public Health; **MHSU**, Maternal Health Services Use; **MoH**, Ministry of Health; **MOR**, Median Odds Ratio; **ODK**, Open Data Kit; **PAB**, Project Advisory Board; **PCV**, Proportional Change in Variance; **PMA-MNH**, Performance Monitoring for Action-Maternal Neonatal Health; **RMNH**, Reproductive, maternal and neonatal health.

95% CI: 1.04 2.99) higher for infants born to mothers who perceived the community did not support postnatal care than for babies born to mothers who believed the community supported it. Infants whose mothers had trouble accessing PNC following the COVID-19 pandemic were dropped out of Pentavalent five times higher (AOR = 5.06, 95% CI: 1.32–19.48). Pentavalent dropout is more than three times (AOR = 3.78, 95% CI: 1.75–8.16) higher among those who lived in the former SNNP region and more than two times (AOR = 2.75, 95% CI: 1.17–6.43) higher among infants born to mothers who lived in a community with a high level of poverty.

## Conclusion and recommendation

The dropout rate from the first to the third pentavalent vaccine is higher than what has been reported in other studies in Ethiopia and abroad. Residing in a high-poverty community, giving birth at home, perceiving that the community did not support postnatal care, and having trouble accessing PNC following COVID 19 significantly determined pentavalent dropout. Improving health education, healthcare access, institutional delivery, and the health systems linkage are therefore essential to addressing these problems in the targeted population.

## Background

Vaccination is a lifetime investment that provides substantial health benefits during all stages of life [1]. Globally, there are an estimated 23 million children younger than one year old who did not receive the recommended basic vaccines, and Ethiopia represents one of the ten countries where nearly 60% of such children reside [2]. The alarming decline in childhood vaccination is leaving millions of children at risk from devastating but preventable diseases [3]. The commonly identified barriers to childhood vaccination encompasses those related to access, health system, concerns and beliefs on vaccination, health perceptions and experiences, knowledge and information, and social or family influences [4]. The maintenance of high vaccination coverage rates is necessary to eradicate vaccine-preventable diseases globally [5]. However, vaccination dropout poses a significant challenge in ensuring that children receive the full protection provided by vaccines. Low dropout rates indicate good accessibility and utilization of vaccination services, which is used as an indicator of the performance of an immunization program [6], which is critical to preventing morbidity and mortality from vaccine preventable diseases [7].

Across the 25 Sub Saharan African countries, inequality in unvaccinated children was disproportionately concentrated among disadvantaged subgroups [8]. In Ethiopia, individual vaccine coverage has improved, but the proportion of fully vaccinated children with all recommended vaccines remains low [9] and children who have not received any vaccinations at all increased from 16% in 2016 to 19% in 2019 [10,11]. Spatiotemporal distributions of vaccination coverage in Ethiopia from 2000 to 2019 found that vaccination coverage in Ethiopia substantially varied across the subnational and local levels [12].

As part of the Ethiopia National Expanded Program on Immunization Comprehensive Multi-Year Plan (2021–2025), it was planned to reduce both Pentavalent 1–3 and Pentavalent 3-MCV1dropout rates to 2% nationally and less than 5% in all districts by 2025. Reducing a high dropout rate in vaccination is also one of the national priorities [13]. Pentavalent vaccine is an important global and national indicator for vaccine dropout and Pentavalent 3 vaccine coverage is a vital indicator for assessing immunization program performance [14]. Since it is now far away from the target, as part of its efforts to ensure that no child is left behind in its immunization program, Ethiopia has started the Re-Ignition of the Big Catch-Up (BCU) initiative by February 2025 [15].

In this study, a GPS coordinates data was used, which can show program managers which locations are not receiving adequate vaccination services, defaulters, provides more accurate denominators of affected population group, and informs what vaccination delivery strategies should be used to optimize coverage and equity. It can also improve monitoring of vaccination programs [16]. This research aimed to identify the level of pentavalent vaccine dropouts during the first year of life and its determinants, which will help to improve child vaccination coverage to close the remaining gaps and inequalities through policy implications.

## Methods

### Study setting and period

PMA-Ethiopia Panel study was conducted in five regions and one city administration that collectively represent 90% of the population in Ethiopia: Addis Ababa, Afar, Amhara, Oromia, SNNP, and Tigray (Fig 1). In this dissertation, the panel data from the three big regions (Amhara, Oromia, and the former SNNP) were used, where all women who were pregnant or less than 6 weeks postpartum at the time of enrollment were followed up through 1-year postpartum. Hence, a cohort of infants born to women enrolled in the fall of 2019 was considered. The survey data was collected between October 2019 and August 2021.

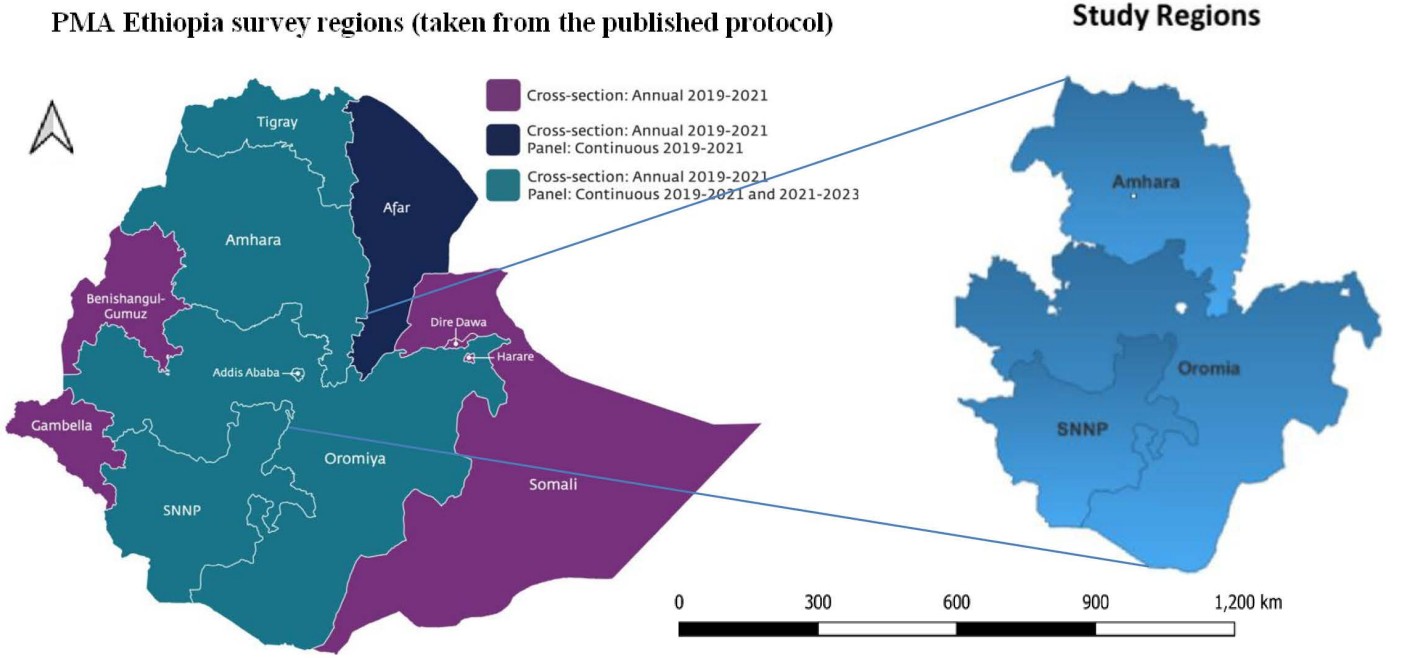

**Fig 1. Study areas map.**

## Study design

PMA Ethiopia panel survey was a longitudinal study conducted among a cohort of pregnant women and their infants through one-year postpartum period.

## Study population

All alive infants born to panel women of reproductive age (15–49 years) who received the first dose of Pentavalent and were living in study regions are population of the study.

**Inclusion criteria.** The women of reproductive age who had been followed up from pregnancy to 1 year postpartum and their alive infants during one-year postpartum follow-up were included in the analysis.

**Exclusion criteria.** Visitors and their infants were not part of the study in PMA Ethiopia. Mothers who were lost to follow-up, declined to participate in a one-year follow-up interview, or who moved permanently will not be included in the analysis of this study.

## Sample size determination

The detailed sample size calculation for PMA Ethiopia are explained elsewhere [17]. We included all 1295 infants available in the dataset in the analysis, as it was calculated that a minimum sample size of 915 was required for this study.

**Sampling techniques.** A multi-stage cluster sampling design with urban-rural stratification was used to draw the enumeration areas (EAs) separately from urban and rural strata within the regions where a probability proportional to size within the strata was used.

## Data collection procedures

Coordinated by regional coordinators and assisted by their supervisors, the female resident enumerators (REs), who were trained in collecting data using smartphone technologies, mapped and listed every household within the EAs to create a sampling frame from which women were screened for eligibility. After census and female screening were completed, all eligible consented women of 15–49 years of age in the EAs were enrolled and provided an identification card (ID) with QR Code that was filled with relevant information. Following the collection of baseline data across a range of gestational ages, from less than one month pregnant to six weeks postpartum, study participants were followed up through one year postpartum when vaccination related data was prospectively collected. This allowed mothers to readily recall pertinent information about infant vaccinations since it was captured timely during the actual follow-ups.

## Data collection tool

All questionnaires were developed to ensure alignment on priority indicators for the Ministry of Health (MoH), Project Advisory Board (PAB), and Bill and Milinda Gate Foundation (BMGF) strategy teams. The 2016 PMA-MNH survey served as a base for the panel survey, with previous PMA2020, the global Performance Monitoring for Action (PMA) surveys, and Demographic and Health (DHS) surveys used as the foundation for cross-sectional measures. Pretest was conducted in Addis Ababa and the surrounding Oromia zones in May 2019. The panel interviews relevant to this study's objective included the use of maternal health services, respectful maternity care during labor and delivery, whether the infant received any vaccine, type of vaccines received, dates on which the respective vaccines were received, maternal socio-demographic factors, community perceptions, geographic variables and others. All the questionnaires were programmed using the Open Data Kit (ODK).

## Dependent variable

Pentavalent vaccine dropout: In this study, pentavalent vaccine was considered to be dropped out when an infant received the first dose (Pentavalent 1) but missed the third dose (Pentavalent 3).

## Independent variables

The independent variables were categorized as level I (individual level variables) and level II (community level variables).

Level 1 (Individual-level factors) included maternal education, wealth quantile, infant birth birth registration at kebele, family size, perception towards community encouragement of PNC, antenatal care, birth place, and difficulty of accessing postnatal care after the covid-19.

Community level variables: In addition to place and region of residence, individual level attributes were aggregated at the community (EA) level to create aggregate community level variables. The aggregate variables were then classified as low or high based on the distribution of the proportion values calculated for each EA. Since the aggregate variables were not normally distributed for all the constructed community variables, median value instead of mean value was used as cut off point for the categorization. Accordingly, Community-level poverty was categorized as high if the proportion of women from the two lowest wealth quintiles in a given community was 21–100% and low if the proportion was 0–20%. Community-level illiteracy was categorized as high if proportion of women who never attended school was 34–100% and as low if it was 0–33%. Community-level media exposure was also categorized as high if the proportion was 39–100% and as low if the proportion of women who use media in the community was 0–38%. This approach was used in many studies [18–20].

## Operational definitions

**Enumeration area.**  A geographic area that has a varying number of households and is adopted from the Ethiopian Statistical Service (ESS).

**Wealth quantile.**  Household wealth was determined by giving scores based on the number and kinds of consumer goods they own, source of drinking water, and type of toilet facilities and flooring materials. Scores were derived using principal component analysis, and then households' wealth index was divided into quintiles according to the wealth score as 'lowest quintile', 'lower quintile', 'middle quintile', 'higher quintile' and 'highest quintile' [21].

**Resident enumerators (REs).**  The PMA Ethiopia data collectors who lived in the enumeration areas during the survey and continuously followed the panel women from pregnancy to one year postpartum.

**Southern nations, nationalities, and people's (SNNP) region.**  The PMA data used in this study was collected prior to the ratification of new regional statehoods. Therefore, the SNNP region in this study refers to the region that incorporates all the newly emerged states.

**Data quality control.**  Before the start of data collection, a series of consecutive and separate trainings were conducted for the baseline and follow up interviews, each followed by field pretests. During data collection, the data management team sent an automated short text message to each RE to remind them of the scheduled follow-up date, the supervisors checked and re-interviewed 10% of interviews per EA with additional on-spot-checks made by the regional coordinators to verify any mismatches. Every day, the data management team downloaded and cleaned the submitted data so that few errors remained by the end of field work.

## Data source

The anonymous micro-datasets that PMA-Ethiopia collects are publicly available on www.pmadata.org in the comma-separated values (csv) text file, Excel, as well as Stata formats.

**Data processing and analysis.**  STATA version 16.1 software was used for data analysis. Sample weight was applied to compensate for unequal probability selection of clusters and women to minimize selection and non-response bias. The collinearity of explanatory variables was checked using variance inflation factors (VIF). We conducted design-based analysis [22] for correcting variances in all descriptive and bivariate analyses for complex survey design of the PMA survey.

Pentavalent vaccine dropout was categorized as a dichotomous variable (Yes/No). Pentavalent1–3 dropout was analyzed by subtracting the total number of infants who received the third dose of Pentavalent from the total number of infants who received Pentavalent1. The analysis considered the hierarchical nature of the data where women/infants are

nested within EAs. Different models, from null model (without explanatory variable) to the final model (with both individual and community level variables) were fitted to identify individual and community-level factors that are associated with the pentavalent dropout. Akaike's Information Criterion (AIC) was used to compare the models, and the smallest value of the information criterion determined best fit model. Finally, statistically significant association was considered at a p-value of < 0.05. Accordingly, multivariable multilevel logistic regression analysis was employed to determine overall factors determining the pentavalent vaccine dropout.

### Building model

Null Model was performed with no explanatory variables to assess the between-cluster variability. Accordingly, the intra-class correlation (ICC) estimation indicated that the data supported the use of a multilevel modeling. The impact of individual-level factors on pentavalent vaccination dropout was investigated in Model I, which included an individual level variables that shown statistical significance in the bivariate multilevel logistic regression analysis. The impact of community level factors on pentavalent vaccine dropouts was investigated in Model II, which included community level variables that showed statistical significance in the bivariate multilevel logistic regression analysis. The factors from the individual level variables were aggregated to create the community level variables. Lastly, taking into account the variables that shown statistically significant associations in models II and III, the third and final model was constructed for the multiple multilevel logistic regression analysis.

### Ethics approval and consent to participate

PMA Ethiopia received ethical approval from Addis Ababa University, College of Health Sciences (AAU/CHS) (Ref: AAUMF 03–008) and the Johns Hopkins University Bloomberg School of Public Health (JHSPH) Institutional Review Board (FWA00000287). Because this study relied on publicly available data with no identifying information, it was reviewed but exempted from needing additional ethics review approval at JHU and Addis Ababa University. Prior to accessing it, every data set was anonymized. We adhere to the PMA Ethiopia data use and sharing policy, which prohibits sharing information with parties other than those involved in this study.

## Results

### Individual level characteristics of study participants

A total of 1295 infants were included in the final analysis. More than a third of the infants' mothers who completed the one-year follow-up were young adults (aged between 15–24 years) and more than three-fourth, 873 (77.74%) were rural residents. About two-fifths of mothers had no formal education and nearly equal proportion of mothers attended primary schools whereas nearly two- in ten mothers had attended secondary and above school. More than three-fourth (37.68%) of the study participants had exposure to either TV or radio.

Additionally, the results showed that over half of the respondents, 799 (56.63%), gave birth at a medical facility, and the majority of respondents, 1098 (83.71%), had ANC follow up for the index pregnancy. Nearly half of the respondents, 655 (48.09%), had four to seven household members. Also, the result indicated that there were approximately equal numbers of boys 647 (50.34%) and girls 648(49.66%) infants in the analysis sample, and 238 (18.45%) of the infants were born first (Table 1).

### Child vaccination dropouts

We analyzed vaccination dropout for 1295 infants who received the first dose of pentavalent and whose mothers completed a one-year postpartum follow-up, in the three regions. A dropout rate from the first to the third dose of pentavalent was 37.48% (31.73%−43.61%). For the first to second dose, the pentavalent dropout rate was 11.27% (8.59% 14.64%), and for the second to third dose, it was 29.74% (24.64%−35.39%) (Fig 2).

**Table 1. Socio-demographic characteristics of study participants in the study regions, PMA Ethiopia 2019- 2021 cohort.**

| Variable(n = 1295) | Frequency | Weighted % |
|---|---|---|
| **Maternal age (1295)** | | |
| 15–24 | 413 | 34.56 |
| 25–34 | 664 | 48.67 |
| 35 and above | 219 | 16.76 |
| **Residence** | | |
| Urban | 422 | 22.26 |
| Rural | 873 | 77.74 |
| **Mother's educational status** | | |
| Never attended | 483 | 39.76 |
| Primary | 519 | 42.00 |
| Secondary and above | 293 | 18.25 |
| **Region** | | |
| Amhara | 393 | 26.33 |
| Oromia | 480 | 48.99 |
| SNNP | 422 | 24.68 |
| **Wealth** | | |
| Lowest quintile | 212 | 18.73 |
| Lower quintile | 221 | 20.01 |
| Middle quintile | 239 | 20.51 |
| Higher quintile | 267 | 21.37 |
| Highest quintile | 356 | 19.38 |
| **Exposure to media (TV/radio)** | | |
| Yes | 539 | 37.68 |
| No | 756 | 62.32 |
| **ANC Attendance** | 1,098 | 83.71 |
| Yes | 197 | 16.29 |
| No | | |
| **Place of Delivery** | 496 | 43.37 |
| Home | 799 | 56.63 |
| Health facility | | |
| **Family size** | | |
| 1–3 | 416 | 31.93 |
| 4–7 | 655 | 48.99 |
| 7 and more | 224 | 19.08 |
| **Sex of infants** | | |
| Male | 647 | 50.34% |
| Female | 648 | 49.66 |
| **Birth order** | | |
| First born | 238 | 18.45 |
| Non-first born | 1,057 | 81.55 |

## Vaccination dropouts by maternal characteristics

Higher pentavalent dropout was observed among infants born to mothers aged 35 and above years (42.48%), never attended school (44.90%), lived in rural residents (44.23%), reside in SNNP (48.43%), had 7 and above family size

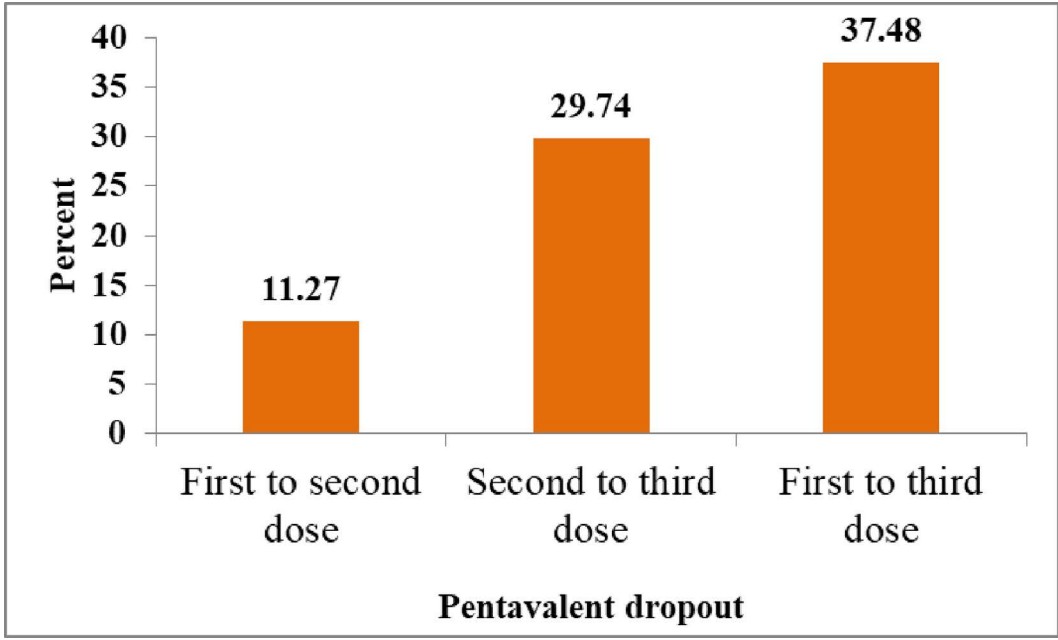

**Fig 2. Pentavalent dropout among a cohort of infants in the study regions, PMA Ethiopia 2019- 2021 cohort.**

(47.45), lowest wealth quintile (50.10%), and have given birth to five and more children before the survey (44.29%). More-over, the result showed that a higher dropout was observed among mothers those did not attend ANC (51.36%), not expo-sure to media (27.45%), did not take at least one shot of tetanus shot during the index pregnancy (46.91%), who gave birth at home (53.54%), and received postnatal care either for the mother or the baby (45.15). A higher dropout (38.98%) was also observed among non-first-born infants (Table 2).

### Determinants of infant vaccination dropouts

To investigate whether pentavalent1–3 dropout is the same or different across clusters, we assessed the extent of intra-class correlation (ICC), which measured the proportion of variances explained by the between-cluster differences, with multilevel logistic regression models. Our null model shows ICC of 0.56, which suggests large heterogeneity in drop-out rates across the clusters. The estimated median odds ratio (MOR), i.e., the unexplained cluster heterogeneity was 6.89, which also supplement that there is high variation between clusters in pentavalent vaccine dropouts. It indicates that if two infants with similar characteristics are randomly selected from different clusters, those from higher-risk clusters have about 7 times odds of experiencing pentavalent dropout compared to infants from lower-risk clusters (Table 3).

### Model comparison and fitness

The result from model I showed that differences in individual level characteristics account for 47% of the variation in pentavalent vaccination dropout, also model II showed that differences amongst communities level characteristics account for 42% of the variation. Model III, the final model indicated that 41% of the variation in vaccination dropout is attributed to both individual and community level differences.

Since we compared models with different numbers of predictor variables, we performed comparison of the goodness of fit of the models using likelihood-ratio test instead of Log-Likelihood. Accordingly, model III was the best-fitted model since it had the lowest AIC value (1253.95) and the highest likelihood ratio (−612.97). The proportional change in variance

**Table 2. Pentavalent vaccine dropouts during infancy by maternal characteristics of the study participants, PMA Ethiopia 2019- 2021 Cohort.**

| Background Characteristics | | Penta1–3 dropout n (weighted %) | Total Number of Children |
|---|---|---|---|
| **Overall** | | **434 (37.48)** | **1295** |
| **Maternal age** | 15–24 | 141 (38.26) | 412 |
| | 25–34 | 208(35.21) | 664 |
| | 35 and above | 85 (42.48) | 219 |
| **Maternal Education** | Never attended | 199 (44.90) | 483 |
| | Primary | 186 (39.05) | 519 |
| | Secondary and above | 49 (17.72) | 293 |
| | | | |
| **Residence** | Urban | 55 (13.91) | 422 |
| | Rural | 379 (44.23) | 873 |
| **Region** | Amhara | 82(23.03) | 394 |
| | Oromia | 164(39.76) | 480 |
| | SNNP | 188(48.43) | 421 |
| **Family Size** | 1–3 | 110(31.109) | 416 |
| | 4–6 | 220(37.70) | 655 |
| | 7+ | 102(47.60) | 224 |
| **Wealth quantile** | Lowest quintile | 104 (50.10) | 212 |
| | Lower quintile | 106 (49.14) | 221 |
| | Middle quintile | 86 (36.19) | 239 |
| | Higher quintile | 98 (40.12) | 267 |
| | Highest quintile | 40(11.72) | 356 |
| **Exposure to media(TV/radio)** | Yes | 122(10.03) | 539 |
| | No | 312 (27.45) | 756 |
| **Mother's TT injection** | Yes | 237(32.01) | 820 |
| | No | 197(46.91) | 475 |
| **Birth order** | First-born | 65(30.87) | 238 |
| | Non-first born | 369(38.98) | 1057 |
| **ANC** | Yes | 340(34.78) | 1098 |
| | No | 94(51.36) | 197 |
| **Place of birth** | Health facility | 180(25.19) | 799 |
| | Home | 253(53.54) | 496 |

(PCV) from model 3 (the full model) shows that both individual and community level factors account for 44% of the odds of Pentavalent dropout (Table 3).

## Determinants of pentavalent dropout

After controlled for the individual and community level factors simultaneously in the final multivariable multilevel logistic regression analysis, only birthplace, residence, maternal perception towards community encouragement of postnatal care, facing difficulty of accessing postnatal care after the covid-19 pandemic, community poverty level, and region of residence were significantly associated with pentavalent vaccination dropout at a p-value of < 0.05. Pentavalent dropout rate was about two times (AOR = 1.84, 95% CI: 1.24–2.72) more likely happened among infants born at home compared to those born at health facility.

**Table 3. Measures of variations and model fit statistics in pentavalent1-3 dropout among the study participants, PMA Ethiopia 2019- 2021 Cohort.**

|  | Null Model | Model 1 | Model 2 | Model 3 |
|---|---|---|---|---|
| **Random effect result** |  |  |  |  |
| ICC | 56% | 47% | 42% | 41% |
| MOR | 6.89 | 5.03 | 4.36 | 4.28 |
| PCV (%) | Ref | 30.42% | 42.40% | 43.79% |
| **Model fit statistics** |  |  |  |  |
| Likelihood ratio test | −656.22 | −620.01 | −632.82 | −612.97 |
| AIC | 1316.45 | 1272.01 | 1281.64 | 1253.95 |
| BIC | 1326.78 | 1354.67 | 1322.97 | 1326.27 |

*AIC : Akakie Information Criterion; ICC , intra cluster correlation coefficient; PCV , proportional change in variance ; MOR , median odds ratio .

Similarly, the findings showed that infants born to mothers who perceived that the community didn't support postnatal care were approximately two times (AOR = 1.77, 95% CI: 1.04–2.99) more likely to have dropped out of pentavalent vaccine than infants born to mothers who perceived the community supported it.. On the other way, compared to those who did not seek care at that time, infants whose mothers had trouble accessing PNC following the COVID-19 pandemic were dropped out of Pentavalent five times higher (AOR = 5.06, 95% CI: 1.32–19.48), whereas the dropout rate was by 38% (AOR = 0.62, 95% CI: 0.42–0.91) less likely happened among mothers who did not experience difficulties.

Moreover the final model analysis showed that infants born to mothers who lived in the community where poverty level was high dropped out of pentavalent vaccine more than two times (AOR = 2.75, 95% CI: 1.17–6.43) compared to those lived in the community where poverty level is low. The infants born to mothers who lived in the former SNNP region were more than three times (AOR = 3.78, 95% CI: 1.75–8.16) dropped out of pentavalent vaccine compared with those from Amhara region.

After adjusted for the individual and community level factors, community level illiteracy, community level media exposure, household wealth quantile, and being from the rural community were not significantly associated with Pentavalent dropout (Table 4).

## Discussions

This study aimed to identify the level of dropouts and its determinants using prospectively collected community level data. The dropout rate between the first and third doses of the pentavalent vaccination, which serves as an important indicator for vaccine dropout, was the main focus of this study.

The result indicated that a dropout from the first to the third dose of Pentavalent was 37.48%, which is significantly higher compared to the pentavalent vaccine dropout rate reported by an earlier study conducted in Ethiopia [23]. The discrepancy could be resulted from the difference in the methods used given that the earlier study was a cross-sectional pocket study and this study was a longitudinal large scale survey. It is also higher than the dropout rate found in another recent study that included Addis Ababa, the nation's capital, and other two other regions [24]. The dropout rate is much higher even when compared with the finding from the Sub-Saharan African countries, which was 20.9% and other studies conducted in Ethiopia [25]. The difference in this case could be resulted from difference in study settings.

Pentavalent dropout rate was about two times(AOR = 1.84, 95% CI: 1.24–2.72) more likely happened among infants born at home compared to those born at health facility which is also supported by the result from demographic and health surveys of SSA countries [25,26]. The possible explanation is that those mothers who delivered at home might have low exposure to the health care system and low awareness about importance of vaccinating infants. Therefore, reduced early

**Table 4. Multivariable multilevel logistic regression results of individual-level and community level factors determining pentavalent vaccine dropout, PMA Ethiopia 2019- 2021 Cohort.**

| Variables | Model I aOR (95% CI) | Model II aOR (95% CI) | Model III aOR (95% CI) |
|---|---|---|---|
| **Individual level factors** | | | |
| **Maternal education (Ref: secondary above)** | | | |
| Never attended | 0.97(0.53-1.76) | | |
| Primary | 0.86(0.50-1.48) | | |
| **Wealth quantile (Ref: lowest)** | | | |
| Lower quintile | 1.28(0.74-2.22) | | 1.28(0.75-2.21) |
| Middle quintile | 0.84(0.48-1.49) | | 0.90(0.51-1.59) |
| Higher quintile | 1.15(0.62-2.13) | | 1.48(0.78-2.80) |
| Highest quintile | 0.29(0.12-0.66)* | | 0.62(0.25-1.55) |
| **Birth registered at Kebele (Ref: Yes)** | | | |
| No | 1.55(0.91-2.64) | | |
| **Antenatal care (Ref: yes)** | | | |
| No | 1.44(0.86-2.41) | | |
| **Birthplace (facility)** | | | |
| Home | 1.85(1.24-2.75)* | | 1.84(1.24-2.72)* |
| **Family size(Ref: 7–10)** | | | |
| 1–3 children | 0.73(0.44-1.23) | | |
| 4–6 children | 0.68(0.43-1.06) | | |
| **Perceived community encouragement of PNC (Ref: Encourage)** | | | |
| Does not encourage | 1.84(1.08-3.15)* | | 1.77(1.04-2.99)* |
| **Difficulty in accessing PNC (Ref: did not seek PNC)** | | | |
| yes | 5.52(1.40-21.81)* | | 5.06(1.32-19.48)* |
| No | 0.63(0.42-0.94)* | | 0.62(0.42-0.91)* |
| **Residence (Ref: Urban)** | | | |
| Rural | | 2.60(1.01-6.71)* | 1.72(0.65-4.56) |
| **Community poverty level (Ref: Low)** | | | |
| High | | 3.67(1.63-8.23)* | 2.75(1.17-6.43)* |
| **Community illiteracy level (Ref: Low)** | | | |
| High | | 0.65(0.14-2.98) | |
| **Community media exposure (Ref: Low)** | | | |
| High | | 2.09(0.43-10.24) | |
| **Region (Ref: Amhara)** | | | |
| Oromia | | 2.75(1.23-6.14)* | 1.99(0.93-4.29) |
| SNNP | | 4.27(1.87-9.75)* | 3.78(1.75-8.16)* |

**\* Statistically significant association at p<0.05.**

contact with health services, low awareness, socioeconomic difficulties, distance, and weak health system linkage might have contributed to higher vaccination dropout rates among home-born infants.

In this study, it was also indicated that infants born to mothers who perceived the community did not support postnatal care were around twice as likely to have defaulted from pentavalent vaccine (AOR = 1.77, 95% CI: 1.04 2.99) as infants born to mothers who perceived the community supported it. This is also inline with another study's finding [27] which showed that who have good perception towards maternal and child services use were 80% less likely to default from complete immunization. This implies that a mother's proper use of health care services, including infant vaccination, will

depend on how she perceives about it. Therefore, awareness creation that targeted the communities is very important to increase the uptake of maternal health services which in turn can influence child vaccination. Also, in the absence of perceived community encouragement, women may not prioritize vaccinations.

On the other way, compared to those who did not seek care, infants whose mothers had trouble accessing PNC following the COVID-19 pandemic were dropped out of Pentavalent approximately three times higher (AOR = 5.06, 95% CI: 1.32–19.48), whereas the dropout rate was by 38% (AOR = 0.62, 95% CI: 0.42–0.91) less likely happened among mothers who did not experience difficulties. The reason for this could be that maternal and child health care are interwoven, meaning that child related services, such as vaccinations, will not be provided if the postpartum mother does not show up at the health facility. This was evidenced by another study's findings [28] which indicates that those did not attend PNC where by 74% less likely completed their child vaccination.

Moreover the final model analysis showed that infants born to mothers who lived in the community where poverty level is high dropped out of Pentavalent vaccine more than two times (AOR = 2.75, 95% CI: 1.17–6.43) compared to those lived in the community where poverty level is low. This finding is consistent with a finding from further analysis of EDHS 2016, which found that children living in community with higher wealth index were twice as likely to receive vaccinations as children living in poor community [29]. This could be because the communities living in poverty may put their immediate survival requirements ahead of immunizations and other preventative health care services. They may relocate frequently in search of employment or better living conditions, which could interfere with their infant vaccinations.

The infants born to mothers who lived in the former SNNP region were more than three times (AOR = 3.78, 95% CI: 1.75–8.16) dropped out of pentavalent vaccine compared with those from Amhara region. This is supported by another study in which significant differences in the odds of vaccination coverage was found among the different regions of Ethiopia [30]. Numerous factors, such as cultural differences, misinformation about vaccines, the accessibility of roads and other health care infrastructures, and the existence of diversities in religions or customs, might have contributed to the regional disparity in vaccination dropout rates, which needs further exploration of the reasons using qualitative approach.

Though they were found to be significantly associated in other studies [24–28,31–35], in this study, factors like maternal education, media exposure, household wealth quantile, ANC attendance, community illiteracy level, and family size were not significantly associated with pentavalent vaccine dropouts.

## Strengths

PMA-Ethiopia represents a significant new source of data for researchers, policy makers, and program implementers trying to improve continuum RMNH care in Ethiopia. The use of design-based analysis, which considered the hierarchical nature of the data, is strength of this study.

## Limitations

The panel respondents are not representative of all women but are representative of 90% of pregnant/recently pregnant women in the three regions.

## Conclusion and recommendation

Dropout from the first to the third dose of pentavalent is higher than the findings from different studies conducted in Ethiopia and abroad. Higher infant pentavalent vaccination dropout rates were significantly determined by a combination of factors, including, giving birth at home, perceiving that the community did not support postnatal care, and having trouble receiving PNC following COVID 19. Improving healthcare access, institutional delivery, health education, and the health system linkages are all important to address these problems. Raising awareness in communities with high rates of poverty is also very crucial for increasing the use of maternal health services, which can have an impact on childhood vaccination.

The reasons why infants dropped from pentavalent vaccine series need to be explored qualitatively from the viewpoints of caregivers and providers.

## Acknowledgments

Our great thank goes to Jimma University, School of Graduate studies, and Department of Population and Family Health for all the whole facilitation of this research work. We would also like to extend our thanks for PMA Ethiopia staffs from the PIs to the resident enumerators, who contributed to PMA Ethiopia panel survey.

## Author contributions

**Conceptualization:** Tariku Tesfaye Bekuma, Saifuddin Ahmed, Muluemebet Abera.

**Data curation:** Tariku Tesfaye Bekuma, Assefa Seme, Muluemebet Abera.

**Formal analysis:** Tariku Tesfaye Bekuma, Assefa Seme, Saifuddin Ahmed, Muluemebet Abera.

**Funding acquisition:** Assefa Seme.

**Investigation:** Tariku Tesfaye Bekuma, Assefa Seme.

**Methodology:** Tariku Tesfaye Bekuma, Assefa Seme, Saifuddin Ahmed, Muluemebet Abera.

**Project administration:** Tariku Tesfaye Bekuma, Assefa Seme.

**Resources:** Assefa Seme.

**Software:** Tariku Tesfaye Bekuma, Saifuddin Ahmed, Muluemebet Abera.

**Supervision:** Tariku Tesfaye Bekuma, Saifuddin Ahmed, Muluemebet Abera.

**Validation:** Tariku Tesfaye Bekuma, Assefa Seme, Saifuddin Ahmed, Muluemebet Abera.

**Visualization:** Tariku Tesfaye Bekuma, Assefa Seme, Saifuddin Ahmed, Muluemebet Abera.

**Writing – original draft:** Tariku Tesfaye Bekuma.

**Writing – review & editing:** Tariku Tesfaye Bekuma, Assefa Seme, Saifuddin Ahmed, Muluemebet Abera.

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
