## [Decision Letter · Decision Letter 0]

Dear Dr. Bekuma,

Thank you for submitting your manuscript to PLOS ONE. After careful consideration, we feel that it has merit but does not fully meet PLOS ONE’s publication criteria as it currently stands. Therefore, we invite you to submit a revised version of the manuscript that addresses the points raised during the review process.

We look forward to receiving your revised manuscript.

Kind regards,

Mohammed Hasen Badeso, Epidemiologist

Academic Editor

PLOS ONE

3.Your ethics statement should only appear in the Methods section of your manuscript. If your ethics statement is written in any section besides the Methods, please delete it from any other section.

4. We notice that your supplementary figures are included in the manuscript file. Please remove them and upload them with the file type 'Supporting Information'. Please ensure that each Supporting Information file has a legend listed in the manuscript after the references list.

Reviewers' comments:

Reviewer's Responses to Questions

**Comments to the Author**

1. Is the manuscript technically sound, and do the data support the conclusions?

Reviewer #1: Yes

Reviewer #2: No

2. Has the statistical analysis been performed appropriately and rigorously?

Reviewer #1: No

Reviewer #2: No

3. Have the authors made all data underlying the findings in their manuscript fully available?

Reviewer #1: Yes

Reviewer #2: No

4. Is the manuscript presented in an intelligible fashion and written in standard English?

Reviewer #1: Yes

Reviewer #2: No

Reviewer #1: 1. Add objective to introduction section than a sub-section in abstract?

2. There are a plenty of research o vaccination drop out in Ethiopia with even with spatial and weighted regression analysis, what you adds?

3. Why you don’t use other random effect statistics, median odds ratio, proportional change in variance

4. I am not sure with your community level variables? What are your references cite it?

5. What is the your rational to do spatial analysis, do you think there is a variation across regions? If so how do you know? Have you done auto-correlation?

6. Don’t compare apple to orange, compare apple to apple in your discussion, why your pocket study compared with sub-Saharan African study?

7. Improve the language, consistency and grammar of your entire text?

Reviewer #2: Reviewer concern

General comments

Throughout the paper there are major editorial and structural issue which requires rearticulating and correction.

This study is not appropriable titled since the the objective of the study is not addressed

The title

The tittle is not related to your research it is not a spatial study since it does not show spatial related methods and findings

Abstract

The background section of the abstract is just copy from your background section as a result please paraphrasing

On conclusion you state higher pentavalent dropouts; what is your baseline to state high?

Your recommendation is not in line with your finding; does you assess whether or not outreach programs gap for the finding?

Background

The first paragraph is just one sentence there for merge with succeeding paragraph

Please maintain the chorological order of information I recommended you to construct from global to local

We are now on 2025; what is this study contribution for Ethiopia National Expanded Program on Immunization Comprehensive Multi-Year Plan (2021-2025); rather find other targets that your study will have a contribution.

The background section generally not informative; and it does not clearly show the known and the gap you want to find out; therefore please clearly show the gap as well as known contributing factors.

Method

Why you used three regions, why not you consider all regions considered in data collection since you want to show spatial distribution; it is recommended to consider all.

If you consider mothers who were lost to follow-up as exclusion criteria; this may underestimate the finding since the mother lost follow up the infant may become defaulter

The details of the sample size is not essential just take the total infants based on your inclusions criteria

Please merge data collection tool and measurement

Please state dependent variable and its operational definition

Please rearticulate data quality section as one paragraph

The first and last paragraph of data processing and analysis is not part of data analysis remove and use for data measurement and operational definition section respectively

Ethics approval and consent to participate; I think it is not applicable since you used secondary data

The method section has not single information about spatial autocorrelation analysis, hot spot analysis, and spatial interpolation; so please clearly right about these issues otherwise it is not a spatial study.

Result

This section is not well articulating please rearticulate all the result section

Related to infant related characteristics which table has the detail of this section

What does mean vaccination dropouts by background characteristics? Please rearticulating appropriately

As I stated in the method section also here in the result section there is no enough analysis done to show spatial distribution. It is not a spatial study else work on autocorrelation analysis, hot spot analysis, and spatial interpolation

Discussion

The discussion is not well articulated and it does not clearly compare with other studies and does not use appropriate justification for discrepancies

Please show the implication for each contributing factors

**Do you want your identity to be public for this peer review?** For information about this choice, including consent withdrawal, please see our Privacy Policy

Reviewer #1: No

Reviewer #2: No

---

## [Author Response · Author response to Decision Letter 1]

13 May 2025

Point by point responses to reviewers ‘comments

We appreciated all the points raised by the reviewers and addressed as described below.

Reviewer 1 comments

1: Add objective to introduction section than a sub-section in abstract?

• Response: Thanks for the comment. Objective is merged into introduction section now.

2: There are a plenty of research o vaccination drop out in Ethiopia with even with spatial and weighted regression analysis, what you adds?

• Response: Indeed, a lot of researches have been done in Ethiopia on vaccine dropout. This study also employed longitudinal data, asking a cohort of mothers about their infants'vaccinations at six weeks, six months, and a year. Because it encompassed the three major regions, it also provided a broader view. Other surveys like EDHS ask retrospectively about vaccination status over the previous two years of the study whereas others are small in scales which are limited within one zone/woreda of the country.

3. Why you don’t use other random effect statistics, median odds ratio, proportional change in variance

• Response: Accepted! Thanks so much for the informative comments. Though we checked it initially, we did present the ICC only. Now, we learnt that it is important to add the median odds ratio and proportional change in variance measures too. They are both added now. Thanks so much, again.

4. I am not sure with your community level variables? What are your references cite it?

• Response: We fully revised the community level variables and we used three references (Michael 2020, Hailay 2024, Kasiye 2021) which are cited as in the document.

5. What is the your rational to do spatial analysis, do you think there is a variation across regions? If so how do you know? Have you done auto-correlation?

• Response: A valid point and we appreciated the comment. Yes, there is significant regional difference and we initially did not intend to show the detailed spatial analysis in this paper as it was not objective of this paper. As a result, we have omitted spatial related statements both from the title and the document. Thanks again for the concern.

6. Don’t compare apple to orange, compare apple to apple in your discussion, why your pocket study compared with sub-Saharan African study?

• Response: Thanks so much for the important feedback. Now, discussion is fully revised as per your kind comments and the changed sessions are shown in TRACKED changes.

7. Improve the language, consistency and grammar of your entire text?

• Response: The document is fully revised from the beginning and all language related issues are well addressed. Thanks so much, for the comment.

Reviewer 2 comments

General comment:

• Throughout the paper there are major editorial and structural issue which requires rearticulating and correction. This study is not appropriable titled since the objective of the study is not addressed

Response: The paper is now thoroughly revised and language related issues are well addressed. The title is modified in a way it better fits to the document. Thanks,

Title

• The tittle is not related to your research it is not a spatial study since it does not show spatial related methods and findings

Response: Yes, we accepted the kind comments. We are using a multilevel analysis with odds ratio and we did not intend to do the detailed spatial analysis. As a result, we removed it both from the title and the content now. Thanks!

Abstract:

• The background section of the abstract is just copy from your background section as a result please paraphrasing. Conclusion you state higher pentavalent dropouts; what is your baseline to state high? Recommendation is not in line with your finding; does you assess whether or not outreach programs gap for the finding?

Responses:

• Background and recommendations are both revised/paraphrased as per the kind comments (shown in tracked changes).

• In conclusion, “Higher” is used to describe the difference among different population sub-groups/a category of explanatory variables included in this study (the variables those significantly affected the dropout). It was not about comparison with other studies’ findings. Thank you for raising the concern.

• Recommendation is revised accordingly. The point is well addressed now.

Introduction:

• The first paragraph is just one sentence there for merge with succeeding paragraph

Response: The 1st and 2nd paragraphs are now merged

• Please maintain the chorological order of information I recommended you to construct from global to local.

Response: Misplaced statements/passages are now re-arranged from global to local. Thanks so much for the constructive feedback.

• We are now on 2025; what is this study contribution for Ethiopia National Expanded Program on Immunization Comprehensive Multi-Year Plan (2021-2025); rather find other targets that your study will have a contribution.

Response: The comment is very important and relevant. Yes, we are already in the target year and the context needs to be changed as a result. Accordingly, we have rephrased the statement to fit the current context and added another target too (the new initiative that Ethiopia recently launched in 2025, Re-Ignition of the Big Catch-Up (BCU)). Thanks,

• The background section generally not informative; and it does not clearly show the known and the gap you want to find out; therefore please clearly show the gap as well as known contributing factors.

Response: The background section is revised all in a way it shows gaps and contributing factors as per the kind suggestion.

Thanks,

Methods section

• Why you used three regions, why not you consider all regions considered in data collection since you want to show spatial distribution; it is recommended to consider all.

Response: We used only the three regions. Because, the panel survey was not conducted across all regions; Addis Ababa is fully urban, Afar is dominantly pastoral whereas panel data collection at Tigray region was interrupted for security reasons. The spatial component is now omitted as it was not the main aim of the study as already stated in the abstract (Objective statement).

• If you consider mothers who were lost to follow-up as exclusion criteria; this may underestimate the finding since the mother lost follow up the infant may become defaulter

Response: Yes, as stated exclusion of those lost to follow up from the data may either over-or-under-estimate the results. To adjust for this, we used a one-year follow-up weight.

• The details of the sample size is not essential just take the total infants based on your inclusions criteria

Response: The details of sample size are now removed as per the kind suggestion.

• Please merge data collection tool and measurement

Response: By measurement, we intended to present the operational measurement of outcome variable, which was generated specifically for this study rather than being initially collected. We have subtitled it now to reflect its true nature.

• Please state dependent variable and its operational definition

Response: This one is addressed in the response to the above comment (data collection tool and measurement). Sorry, it was confusing as it initially stands. Thanks for the constructive comment.

• Please rearticulate data quality section as one paragraph

Response: Thanks so much, for the comment. Data quality section is merged as a single passage now.

• The first and last paragraph of data processing and analysis is not part of data analysis remove and use for data measurement and operational definition section respectively

Response: Thanks! This is addressed accordingly and can be checked in the tracked changes.

• Ethics approval and consent to participate; I think it is not applicable since you used secondary data

Response: Ethics approval and consent to participate, which was stated in the paper refers to the original PMA Ethiopia survey. We wanted to provide evidence showing that the original survey was ethically cleared (as used in many studies). It was also separately requested during the online manuscript submission to the journal. Thanks again!

• The method section has no single information about spatial autocorrelation analysis, hot spot analysis, and spatial interpolation; so please clearly right about these issues otherwise it is not a spatial study.

Response: Thanks so much! We accepted your kind comments. Yes, it’s not a spatial study. As already stated in the abstract section, the study aimed to examine the level of pentavalent vaccine dropouts and determinants during infancy. Spatial analysis is already underway in another paper and we removed spatial statements both from the title and the content.

Thanks again,

Results Section

• This section is not well articulating please rearticulate all the result section

Related to infant related characteristics which table has the detail of this section

What does mean vaccination dropouts by background characteristics? Please rearticulating appropriately

Response: Result section is thoroughly revised and all the necessary changes are made. Infant related characteristics are now merged as study participants ‘characteristics which was originally stated only in a passage. Background characteristics were to mean maternal different characteristics. It’s now modified accordingly. Thanks,

• As I stated in the method section also here in the result section there is no enough analysis done to show spatial distribution. It is not a spatial study else work on autocorrelation analysis, hot spot analysis, and spatial interpolation

Response: Well appreciated! As responded above in methods session, the study aimed to examine the level of pentavalent vaccine dropouts and determinants during infancy. As a result, the spatial related statements are omitted for the stated reason.

Discussions session

• The discussion is not well articulated and it does not clearly compare with other studies and does not use appropriate justification for discrepancies

Please show the implication for each contributing factors

Response: Discussion is now thoroughly revised and sufficient justifications are added for the discrepancies. The implications for each factor are also explained well. Thanks so much for the informative comments!

---

## [Editor Report · Decision Letter 1]

Level and determinants of pentavalent vaccine dropout during infancy: a hierarchical analysis of community-level longitudinal study

PONE-D-25-04328R1

Dear Dr. Bekuma,

We’re pleased to inform you that your manuscript has been judged scientifically suitable for publication and will be formally accepted for publication once it meets all outstanding technical requirements.

Kind regards,

Mohammed Hasen Badeso, Epidemiologist

Academic Editor

PLOS ONE
---

## [Editor Report · Acceptance letter]

PONE-D-25-04328R1

PLOS ONE

Dear Dr. Bekuma,

I'm pleased to inform you that your manuscript has been deemed suitable for publication in PLOS ONE. Congratulations! Your manuscript is now being handed over to our production team.

Kind regards,

on behalf of

Mr Mohammed Hasen Badeso

Academic Editor

PLOS ONE